# Mechanical and Biological Evaluation of Melt-Electrowritten Polycaprolactone Scaffolds for Acetabular Labrum Restoration

**DOI:** 10.3390/cells11213450

**Published:** 2022-10-31

**Authors:** Matthias X. T. Santschi, Stephanie Huber, Jan Bujalka, Nouara Imhof, Michael Leunig, Stephen J. Ferguson

**Affiliations:** 1Institute for Biomechanics, Department of Health Sciences and Technology, ETH Zurich, 8092 Zürich, Switzerland; 2Department of Orthopaedic Surgery, Schulthess Clinic, 8008 Zurich, Switzerland

**Keywords:** acetabular labrum, melt electrowriting, biofabrication

## Abstract

Repair or reconstruction of a degenerated or injured acetabular labrum is essential to the stability and health of the hip joint. Current methods for restoration fail to reproduce the structure and mechanical properties of the labrum. In this study, we characterized the structure and tensile mechanical properties of melt-electrowritten polycaprolactone scaffolds of varying architectures and assessed the labrum cell compatibility of selected graft candidates. Cell compatibility was assessed using immunofluorescence of the actin skeleton. First, labrum explants were co-cultured with scaffold specimen to investigate the scaffold compatibility with primary cells. Second, effects of pore size on pre-cultured seeded labrum cells were studied. Third, cell compatibility under dynamic stretching was examined. Grid-like structures showed favorable tensile properties with decreasing fibre spacing. Young’s moduli ranging from 2.33 ± 0.34 to 13.36 ± 2.59 MPa were measured across all structures. Primary labrum cells were able to migrate from co-cultured labrum tissue specimens into the scaffold and grow in vitro. Incorporation of small-diameter-fibre and interfibre spacing improved cell distribution and cell spreading, whereas mechanical properties were only marginally affected. Wave-patterned constructs reproduced the non-linear elastic behaviour of native labrum tissue and, therefore, allowed for physiological cyclic tensile strain but showed decreased cell compatibility under dynamic loading. In conclusion, melt-electrowritten polycaprolactone scaffolds are promising candidates for labral grafts; however, further development is required to improve both the mechanical and biological compatibility.

## 1. Introduction

The hip joint is a ball-and-socket joint consisting of the femoral head (ball) and the acetabulum (socket). A fibrocartilaginous ring, the acetabular labrum, is attached to the acetabular rim and provides an essential stabilization and sealing function to the hip joint [1,2,3]. Loss of function of the labrum can result in hip joint instability and cartilage degeneration [1,2,3,4,5] and contribute to the development of secondary osteoarthritis [6,7,8]. Therefore, repair or reconstruction of the labral tissue is essential to restore biomechanical function and to preserve the hip joint. Today, autograft and allograft tissues are clinically used for reconstruction [9], both with their own benefits and shortcomings. Autografts can result in donor site morbidity, whereas allografts offer an off-the-shelf solution but carry a small risk of rejection and disease transmission [10]. Reported graft choices include ligament, muscle, and tendon tissues [11,12,13,14,15], which fail to adequately reproduce the distinct form, structure, and mechanical properties of the native labrum tissue. Advanced 3D printing techniques such as melt-electrowriting have the potential to overcome these shortcomings and offer promising options for future graft choices.

Material properties of musculoskeletal tissues and their biomechanical functions are closely related. Successful restoration, therefore, requires matching the mechanical properties of the graft to those of the native tissue. The articulating surface, which interacts directly with the articular cartilage of the femoral head, is coated by lubricating molecules (lubricin) which enable low-friction surface motion [16,17]. The deeper and outer tissue portion of the labrum is composed of highly aligned circumferentially running collagen fibre bundles, which provide exceptional circumferential tensile strength when compared with meniscus or cartilage tissues [18]. Tensile-to-break testing of bovine labrum specimens resulted in a stress–strain curve which contains a non-linear toe region, a subsequent near-linear region which is used to evaluate the tissue stiffness, and at higher strains, a short yield region with decreasing tangent modulus prior to abrupt tissue failure. On average, the non-linear toe region extended to 10.3 ± 3.3%, with a tangent modulus of 5.83 ± 4.11 MPa and the near-linear region up to the yield point at 19.5 ± 4.2%, with a tangent modulus of 74.7 ± 44.3 Mpa; tissue failure occurred at a maximum strain of 26.5 ± 7.6% and a maximum stress of 11.9 ± 6.1 MPa [18]. The tensile modulus (stiffness) of human labrum tissue is reported in a similar range, between 24.7 ± 10.8 MPa and 66.4 MPa ± 42.2 [19,20]. Due to the high stiffness in tension and the low fluid permeability (4.98 × 10^−6^ m^4^/Ns ± 3.43), the labrum tissue shows high resistance to interstitial flow, higher than cartilage or meniscus [21]. In summary, these tissue properties allow the labrum to sustain local forces and to act as a seal of the hip joint. In the unloaded hip joint, a negative pressure is created in the articular space to stabilize the hip joint, whereas in the loaded hip joint the synovial fluid is prevented from escaping the intra-articular space and is pressurized, which increases joint lubrication and distributes acting forces more evenly across the articular cartilage [1,2,3]. The articular cartilage is a load-bearing structure and “designed” to withstand high compressive forces, whereas the labrum does not substantially contribute to load bearing in a normal joint [21]. This is reflected in the compressive modulus of the labrum tissue, which at 0.157 ± 0.057 MPa is approximately one-quarter to one-half that of the adjoining acetabular cartilage [18].

The hip joint experiences high cyclic compressive loads during daily activities [22]. Forces acting on the labrum tissue are, therefore, of a dynamic nature, which is not fully considered with single load cycle testing. The mechanical properties of the labrum tissue under cyclic loading have, to our knowledge, not been investigated so far, resulting in incomplete knowledge about the elastic and plastic response of the labrum tissue. However, remarkable strains are repeatedly generated in the labrum tissue during physiological movements of the hip joint, without causing any permanent (plastic) deformations. In the anterior labrum, an average tensile strain of 5% in both the axial and circumferential direction as well as a maximum axial strain of 13.6% and a maximum circumferential strain of 8.4% were measured in human hip cadavers for 12 physiological hip joint positions, axially loaded with 445 N [23]. A subsequent study provided comprehensive data of circumferential strain for the whole labrum and through the complete range of motion of the hip joint (36 different positions). The study confirmed that local strains vary with hip joint position and identified the greatest circumferential strains in the posterior labrum [24]. Based on these observations, we believe that the elastic properties of the labrum may be crucial to maintaining contact between the femoral head and the labrum over the full range of motion, without compromising the physiological movement of the hip joint. Potential labral grafts should, therefore, additionally be evaluated under cyclic mechanical loading and allow for a similar temporary deformation (elastic) as the native tissue under joint-mimicking loading.

In the recent years, melt electrowriting (MEW) has been established as an additive manufacturing process in tissue engineering (TE), which allows the fabrication of porous microfibre constructs from thermoplastic polymers [25]. Many different polymers have been used for MEW applications, and poly(ε-caprolactone) (PCL) has become the gold standard due to its low melting temperature and high biocompatibility [26]. In contrast to electrospinning, where a polymer is solubilized in a suitable solvent and processed to a mostly random micro- or nanofibrous mat, MEW allows the production of highly ordered microfibre structures without the use of potentially toxic solvents [25,27]. Briefly, a setup for MEW requires a head for melting and extrusion through a nozzle, as well as a collector. A polymer is heated and kept molten at a temperature above its melting temperature. The polymer melt is then extruded through a nozzle, which acts as an electrode for high voltage. The collector, acting as a second electrode, creates an electric field, which stabilizes the polymer jet drawn from the nozzle. To be able to deposit the fibres in a writing-like manner, the head and the collector must be able to translate in the x and y directions relative to each other. The translational speed required to draw a straight fibre is called the critical translational speed (CTS). Provided the translational speed is above the CTS, the fibre is dragged behind and will lie down as a straight fibre. Translational speeds below CTS on the other hand will result in deposition of crimped fibres [28]. Process parameters such as melt temperature, polymer flow, nozzle size, high voltage, distance, and translational speed have to be chosen with respect to obtaining and maintaining a stable flow to the nozzle, as well as from the nozzle towards the collector [29]. Fibre diameter produced by MEW, in the lower micrometer range, are highly adjustable and are mainly subject to polymer flow rate and mechanical stretch through the translational motion [30].

The possibility to manufacture highly ordered and porous polymer microfibre structures without the use of solvents has raised interest in conducting further research in a wide range of applications of MEW in TE. In cardiovascular research, Castilho et al. developed a heart patch made of hexagonal MEW microstructrues which allows maturation of stem cell derived cardiomyocytes [31]. In another study, MEW was used for the development of biologically inspired scaffolds for heart valve tissue engineering [32]. MEW has also found its place in the field of TE of musculoskeletal tissues. In 2021, Meng et al. used MEW to produce high precision PLLA scaffolds and studied the in vitro potential for bone tissue engineering [33]. Recently, Dufour et al. published their approach of combining bioprinting and MEW to engineer structurally organized articular cartilage constructs [34]. In another study, the effect of MEW fibre orientation on cellular organization in the context of anterior cruciate ligament engineering was studied [35]. In an attempt to recreate three-dimensional scaffolds, Korpershoek et al. produced structures in a shape similar to the native meniscus [36]. Outside of musculoskeletal and cardiovascular tissues, various studies have investigated the application of MEW structures for other TE purposes, such as skin regeneration and corneal stroma engineering [37,38]. Despite the broad exploration of MEW applications in regenerative medicine, to our knowledge, the potential of MEW scaffolds has not been investigated for labrum restoration. In a study published in 2021, Tey-Pons et al. assessed the potential of polyurethane scaffolds, originally developed and clinically available for symptomatic meniscal tissue restoration. Labral reconstruction was performed in three patients with femoral acetabular impingement which resulted in reduced pain and an improved hip joint function as assessed 4 years after implantation [39]. Anindyajati and colleagues [40] assessed the potential of a combined ultra-high molecular weight polyethylene (UHMWPE) fabric with ultrafine electrospun PCL, the latter part a technique similar to MEW and a material used in the present study. This study revealed more than four times greater resistance to tensile loading than the native labrum tissue and, therefore, requires further development to achieve matching mechanical properties. In addition, although the UHMWPE-PCL composite provided a viable environment in vitro, it must be noted that mouse fibroblast cell lines were used for the assessment of cell compatibility, which might differ from primary labral fibrochondrocytes [41].

The goal of this study was to compare melt-electrowritten PCL scaffolds of varying architecture with respect to the most relevant mechanical properties and identify ideal candidates for labrum reconstruction. Moreover, a preliminary biological evaluation was conducted to provide insights on primary labrum cell growth. We hypothesized that: (i) mechanical properties improve with decreased fibre spacing are independent of construct thickness, (ii) hierarchical scaffolds improve cell spreading without altering mechanical properties, and (iii) biomimetic wave-patterned scaffolds reproduce the non-linear mechanical response of labrum tissue and enhance cell response.

## 2. Materials and Methods

### 2.1. Overview

In this study, PCL constructs with a range of architectures were manufactured using MEW. The fabricated constructs were structurally assessed using scanning electron microscopy, mechanically by means of quasistatic and cyclic uniaxial tensile testing and biologically using static and dynamic cell culture with primary bovine acetabular labrum cells as well as static tissue co-culture with primary bovine acetabular labrum explants. The experiments were conducted at the Laboratory for Orthopedic Technology of ETH Zurich over a time frame of six months.

### 2.2. Material for MEW

Medical grade poly (ε-caprolactone) (PCL) (PURASORB^®^ PC 12, Corbion Inc., Amsterdam, The Netherlands) with M_w_ = 70 kDa was used for the fabrication of all scaffolds. Stored at −20 °C, the aliquots were pre-heated to 80°C for at least 48 h to achieve a uniform melt.

### 2.3. Melt Electrowriting Setup

All scaffolds were fabricated using a custom-built, in-house melt electrowriting device. The device consists of a vertically moving melt head and a horizontally translating collector. The polymer in the melt head was kept in a 5 mL syringe (Braun, Melsungen, Germany) connected to a 22 G blunt needle (Nordson, Westlake, OH, USA). The polymer was kept molten in a heating jacket. Air pressure was applied to extrude the polymer melt through the needle. The collector consisted of a detachable, 1 mm thick, soda–lime glass plate on top of an aluminum plate. Attachment of the glass plate to the aluminum was facilitated by application of negative pressure between the plates. Positive and negative high voltage was applied to the needle and the aluminum plate, respectively. The printing process was controlled through Mach4 (Version 4.2.0.4612, Newfangled Solutions, Livermore Falls, ME, USA), a software commonly used for CNC machines. Furthermore, a climate control chamber (Parameter, Black Mountain, NC, USA) enabled the control of temperature and relative humidity. A custom-made Python tool was used to generate the G-codes for motion commands of the device, enabling production of scaffolds with predefined architectures.

### 2.4. Structure Design and Fabrication

Two main types of structures were manufactured. The first consists of melt-electrowritten PCL fibres that were deposited in different orientations of straight fibre paths. The second structure is made of alternating layers of straight fibres in one direction and waved fibres in the orthogonal direction. For the first type, a total of seven different structures were manufactured with different fibre spacings or densities (see Table 1). For the second structure type, a waved pathway was introduced for the translating collector with a wavelength of 2 mm and amplitude of 500 µm. For scaffold fabrication, the PCL was kept at a melt head temperature of 80 °C. An air pressure of 1 bar was applied to extrude the polymer melt. A positive high voltage of 3.8 kV and a negative high voltage of −1.5 kV were applied to the needle and the collector, respectively. The distance between the tip of the needle and the collector was set to 3 mm. Both high voltage and distance between the tip of the needle and the collector were increased by 10 V and 20 µm for each layer, to compensate for the build height of the structure. The climate chamber was used to keep the environment at 21 °C and 41% relative humidity. For most fibres, the translating speed of the collector was chosen to be 1.25× CTS. The only exception were the thin fibres in the multiscale approach, which were achieved by increasing the translating speed to 20× CTS and, therefore, increasing the mechanical stretch to which the fibres were subjected.

### 2.5. Structural and Material Analysis

Structural analysis of all sample types (*n* = 6 per type) was performed using a scanning electron microscope (JSM-7100F, JEOL, Japan or SU5000, Hitachi, Japan) at the Scientific Center for Optical and Electron Microscopy (ScopeM), Zurich, Switzerland. Biopsy punches (Ø 8 mm) were used to punch out a part of the scaffold. These representative parts were subsequently mounted on SEM specimen stubs (Ø 10 mm) using a conductive carbon tape. Additionally, a platinum/palladium (PtPd) coating of 10 nm thickness was applied to ensure conductivity of the sample. Each structure was examined regarding fibre spacings and fibre thickness using ImageJ v1.53c. Fibre diameters and spacings were measured at the top two layers of the scaffold at 20 different spots per scaffold. With respect to the thin fibres in multiscale structures, only fibre diameter was measured. In addition, the wavelength as well as the amplitude was measured for the waved structures. To detect potential alterations of the PCL through the process of MEW, ATR-FTIR spectra were collected. ATR-FTIR spectra of the MEW PCL constructs were obtained via a Varian 640 Fourier Transform Infra Red Spectrometer (Agilent Technologies, Santa Clara, CA, USA), equipped with a Golden Gate-diamond ATR. All ATR-FTIR spectra were collected with 64 scans in the 400–4000 cm^−1^ spectral region at 4 cm^−1^ resolution. After obtaining the raw spectra, a baseline correction based on the 2nd derivative method (zeroes) was performed with the software of OriginLab (OriginLab Corporation, Northampton, MA, USA).

### 2.6. Mechanical Characterization

The scaffolds were cut into two 35 × 15 mm pieces (*n* = 5) using a cutting tool with microtome blades for quasistatic and cyclic tensile testing. The rectangular sample shape may be more appropriate for polymers and potentially reduces edge effects, as suggested in the Standard Guide for Characterizing Fibre-Based Constructs for Tissue-Engineered Medical Products (ASTM F3510-21). The cut samples were clamped into the mechanical testing device using custom clamps. All mechanical tests were performed with a dynamic material test machine (Instron ElectroPuls^TM^ E10000, High Wycombe, UK), equipped with a 1 kN/25 Nm in-line load cell and an additional 10 N sensor (KD24s, Transmetra, Flurlingen, Switzerland) in series for better signal quality. Additionally, an Epsilon ONE^®^ video extensometer (Epsilon Technology Corp., Jacksonville, WY, USA) was used to track the displacement of the structure between two markings applied on the samples. For the quasistatic tensile test, the samples were stretched to 100% at a strain rate of 0.01 mm/s to derive tensile properties, such as tensile (Young’s) modulus, yield strength, yield strain, and ultimate tensile strength. For the cyclic tensile test of the samples consisting only of straight fibres, a stepwise increase in strain from 0.2% up to 40% at 1 Hz was applied, with 20 cycles per strain to learn more about the elastic behaviour of the samples. In the case of waved structures, a long-term cyclic test was performed at 1 Hz and 10% strain for 1 h.

### 2.7. Biological Characterization

To evaluate the effect of incorporation of fibres with smaller diameter and spacing (multiscale) in grid-like scaffolds with greater (250 µm) fibre spacing, cylindrical scaffold punches (Ø 8 mm, 40 layers) of both types (multiscale and 250/40) were created and prepared for cell culture experiments. To allow for cell attachment, Arg-Gly-Asp peptides were covalently bound to the material surface. Briefly, the hydrophobic scaffolds were incubated in a series of decreasing ethanol concentrations to allow penetration of aqueous solutions. In a next step, they were immersed in 4 M NaOH for 4 h., in 50 mM/50 mM EDC/NHS solution in MES buffer at a pH 5.7 for 1 h (M3671, Sigma Aldrich, St. Louis, MO, USA) and in 1 mg/mL Arg-Gly-Asp (A8052, Sigma Aldrich) in PBS for 2 h. Thorough washing was performed after each incubation step. Scaffolds were sterilized using 70% ethanol and UV light application. Cylindrical labrum tissue specimen (Ø 8 mm, 4 mm thickness) from the superior labrum were collected from bovine hip joints (N = 4; male, between 484 and 903 days) and placed on the scaffolds with the articulating surface facing towards the scaffold. Low serum culture medium (DMEM/F-12, 2% FCS; 1% ITS-G, 1% AA) was added and frequently changed until after 6 weeks of co-culture when the tissue specimen was removed and the cell-containing scaffold investigated. Cells were fixated in 4% formaldehyde for 15 min, permeabilized in 0.1% Triton X-100 (9036-19-5, PanReac AppliChem, Darmstadt, Germany) for 10 min, non-specific binding blocked in 1% bovine serum albumin (A9647, Sigma Aldrich) for 1 h, before f-actin was stained by applying DAPI and Phalloidin Alexa Fluor 568, following manufacturers recommendations (A12380, Thermo Fisher, Reinachm, Switzerland). Samples were imaged and maximum z projections created of three locations of each scaffold side using a confocal microscope with a 10× and 20× magnification objective (Leica DMI8-CS with Leica LAS X SP8 Version 1.0, Wetzlar, Germany) at recommended wavelengths.

In addition to the long-term co-culture experiment, a controlled number of labrum cells were homogenously seeded onto the same types of scaffolds and cell distribution, and morphology was studied as described above. For cell isolation, a portion of the superior labrum from bovine hip joints (N = 6; male, between 485 and 709 days) was collected, minced into tissue fragments, digested in 0.4 % pronase (53702-250KU, Merck Millipore, Billerica, MA, USA) for 2 h and in 0.3 % collagenase type II (17101015, Gibco, Carlsbad, CA, USA) for 18–20 h at 37 °C and a cell suspension obtained by passing through a 100 µm cell strainer. At passage three, 40,000 cells were seeded to each scaffold using a small cell suspension volume of 30 µL and allowed to attach for 20 min before adding more culture medium (DMEM/F-12, 4% FCS; 1% ITS-G, 1% AA) to fully submerge the scaffolds. After 2 days of culture, cells were fixated, stained, and imaged as described above.

To assess the effects of cyclic tensile strain on cell compatibility, 1.5 × 4 mm strips of wave-patterned membranes (waved500) were created, and PCL (704105) handles melted onto both ends using a custom-made mold at 60 °C. To allow for cell attachment, Arg-Gly-Asp peptides were covalently bound to the material surface as described above. Prior to cell seeding, scaffolds were mounted to a commercial stretch chamber (STB-CH-1.5, STREX).

Bovine labrum cells were isolated as described above. At passage three, 300,000 cells were seeded to each scaffold in a high cell density suspension (7500 cells/μL) and allowed to attach for 20 min before the chamber was filled with more culture medium (DMEM/F-12, 4% FCS; 1% ITS-G, 1% AA) to fully submerge the scaffolds. After 3 days of culture, chambers were assigned to two groups: static culture and physiological long-term stretching (10% cyclic tensile strain at 1 Hz for 1 h/day for 5 days) using the STREX system (STB-140-10). Twenty-four hours after the last loading cycle, metabolic activity, cell morphology, or gene expression was investigated. To assess metabolic activity, the cells were incubated with PrestoBlue™ HS Cell Viability Reagent (P50200, Thermo Fisher Scientific, Basel, Switzerland), diluted 1:10 in culture medium for 1.5 h, and fluorescence intensity of the collected supernatant was measured using a plate reader (Infinite M200 PRO, TECAN) at an excitation/emission wavelength of 560/590 nm. To assess growth, cells were fixated, and f-actin was stained as described above. Samples were imaged using fluorescence illumination of the X-Cite 120Q (Lumen Dynamics, Mississauga, ON, Canada), a microscope (IX51, Olympus, Shinjuku, Japan) with the digital microscope camera DP74 (Olympus, Kyoto, Japan), and the cellSens (Standard Version 3.2, Olympus, Japan) software allowing for panoramic imaging of big samples.

### 2.8. Data Analysis

The obtained data were processed with Jupyter notebooks (Python 3.8.8). All statistical analyses and graphs were obtained with GraphPad Prism (v3.2.0) and RStudio (2021.09.0, Build 351). Every tested structure was analyzed regarding Young’s modulus, yield strength, yield strain, and ultimate strength. Multiple linear regression was performed to understand whether fibre spacing and number of layers significantly influence mechanical properties of the structures, with no stepwise selection of the variables (enter method). A log-log transform was performed to counter homoscedasticity. Multiscale structures were excluded from the regression fit due to an insufficient amount of data sets. Additionally, an unpaired t-test was performed to compare means of all examined mechanical properties between 40-layer/250 μm and multiscale scaffolds.

For the statistical evaluation of the metabolic activity assay, a two-sided Wilcoxon matched pairs signed rank test was performed at a statistical significance level of 0.05 using GraphPad Prism 8 (GraphPad Software Inc., La Jolla, CA, USA). The effect size is reported as the average fold change ± standard deviation.

## 3. Results

### 3.1. Structural and Material Characterization

Overall fibre diameter was 17.00 ± 1.27 µm. For the samples 1000/20, 1000/40, 500/20, 500/40, 250/20, 250/40, and multiscale, the mean fibre spacings of the grid were found to be 966.4 ± 10.99 µm, 962.7 ± 15.61 µm, 484.5 ± 8.55 µm, 479.6 ± 12.01 µm, 242.0 ± 18.11 µm, 242.5 ± 25.48 µm, and 237.7 ± 31.41 µm, respectively. The mean fibre diameter of the waved MEW structure was 17.5 ± 1.5 µm. The effective amplitude and wavelength were 260 ± 41 µm and 1.99 ± 0.05 mm, respectively. A representation of the produced structures can be found in Figure 1. 

ATR-FTIR spectra (Figure 2) showed characteristic absorption peaks of the PCL, such as the symmetric and asymmetric stretching of methylene groups (2865 and 2944 cm^−1^), stretching of the ester carbonyl (O-C = O) at 1726 cm^−1^, C-O stretching bands of the crystalline phase of PCL (1294 cm^−1^), and symmetric and asymmetric stretching of the OC-O and C-O-C bonds, at 1189 and 1242 cm^−1^, respectively.

### 3.2. Mechanical Characterization

A graphical representation of the mechanical tests can be found in Figure 3. The values for Young’s modulus increase with decreasing fibre spacing. Multiple linear regression (R^2^= 0.8527, F(2, 27) = 84.91, *p* = 2.256 × 10^−12^) showed that, at alpha level (α = 0.05), fibre spacing has a significant effect on Young’s modulus (*p* = 4.06 × 10^−13^), while the number of layers did not show a significant effect (*p* = 0.242). The unpaired *t*-test between 40-layer/250 µm and multiscale structures did not show a significant difference (α = 0.05, *p* = 0.0537) between the means of Young’s modulus values.

The highest stresses were observed in multiscale specimens. Multiple regression analysis (R^2^ = 0.9518, F(2, 27) = 287.6, *p* = < 2.2 × 10^−16^) showed a significant influence (*p* = <2.2 × 10^−16^) of fibre spacing on ultimate stress. As for Young’s modulus, the unpaired *t*-test did not reveal a significant difference between 40-layer/250 µm and multiscale scaffolds (α = 0.05, *p* = 0.481) regarding ultimate stress.

Multiple regression analysis (R^2^ = 0.8813, F(2, 27) = 108.7, *p* = 1.215 × 10^−13^) showed a significant effect of both fibre spacing (*p* = 5.02 × 10^−14^) and number of layers (*p* = 0.000382) on yield stress. Equal analysis regarding yield strain yielded a worse fit (R^2^ = 0.4783, F(2, 27) = 14.29, *p* = 5.838 × 10^−5^) while still showing significant influence of both fibre spacing (*p* = 7.91 × 10^−5^) and number of layers (*p* = 0.01336) on yield strain. In contrast to Young’s modulus and ultimate stress, the unpaired t-test showed a significant difference between 40-layer/250 µm and multiscale sample means regarding yield stress (*p* = 0.0116). In terms of yield strain, no significant difference was observed (*p* = 0.4802). The values for Young’s modulus, ultimate strength, yield stress, and yield strain are listed in Table 2. Tensile testing of the waved structure revealed a tensile modulus of 0.89 ± 0.15 MPa. The non-linear toe region transitioned to the linear elastic region at a strain of 14.2 ± 1.8%, while plastic deformation was visible at a strain of 39.4 ± 3.5%. Cyclic testing of 3600 cycles resulted in a slightly reduced stiffness.

### 3.3. Biological Characterization

PCL melt-electrowritten scaffolds were compatible with both pre-cultured labrum cells (Figure 4c,d) and primary labrum cells (Figure 4f,g). On both scaffold types, grid-like scaffolds (Figure 4a,c,f) and multiscale scaffolds (Figure 4b,d,g), cell–material interaction was observed. A wider variety of cell shapes was observed on multiscale scaffolds, which allowed the cells to occupy the space between the larger fibres (Figure 4d,g).

Labrum cell compatibility of wave-patterned scaffolds was additionally assessed under dynamic mechanical loading. The overall metabolic activity, a function of cell number and single cell metabolic activity, was reduced significantly to 52% ± 21% in stretched samples compared with static controls (*p* = 0.03, sample size 6). The cells were not able to bridge the gaps and grew along the PCL fibres (Figure 5b–e). A decreased actin signal could be observed in stretched compared to static samples (Figure 5b,e). No consistent change in cell morphology could be identified between stretched and static samples (Figure 5c,d), although cell number seemed to have decreased in some cases.

## 4. Discussion

Overall, the structural characterization revealed mean fibre diameters of 17.00 ± 1.27 µm and 3.47 ± 0.60 µm for thick and thin fibres, indicating a consistent and stable manufacturing process. The variation can be explained by the dynamic nature of the CTS. It has been observed that the CTS decreases over the course of the 7 days of usage, before being replaced by a new batch of polymer. The decreasing CTS, especially during long prints, results in a translational speed of the collector being higher with respect to the current CTS, hence creating higher mechanical drag. A recently published study, a multi-week experiment for the assessment of thermal stability of medical-grade PCL, reported a similar observation of decreasing CTS over the course of the first week. In addition, this study highlighted a time window with stable CTS starting roughly after one week of heating time [42]. This suggests that introducing a pre-heating period could further improve the consistency and quality of the MEW structures. Increasing the translational collector speed from 1.25xCTS to 20xCTS showed a drastic reduction of fibre diameter due to the increased mechanical stretch. Similar observations had been already reported in a study by Hrynevich et al. [30]. ATR-FTIR spectra have revealed similar peaks associated to PCL before and, therefore, hint that no alterations have been made to the PCL through the process of MEW [43].

Uniaxial tensile tests showed that the fibre spacing plays a crucial role for the mechanical properties of the manufactured structures. Multiple linear regression showed a significant effect of the spacing on Young’s modulus, ultimate strength, and yield stress and strain. The number of layers did not influence the tensile properties, except the yield point. In terms of yield stress, the values for 20-layer were higher compared with 40-layer specimens, but only for fibre spacings of 500 and 1000 μm. An explanation for this phenomenon might be the amount of sagging and bridging between fibre crossings. In 20-layer structures, the peaks were smaller, or in other words, the valleys were not as deep compared with 40-layer scaffolds. Due to this structural difference, a higher proportion of fibres were extended in 20-layer structures, ultimately causing higher stresses. This effect did not persist in 250 μm structures, as the small spacing substantially reduces fibre sagging. Furthermore, the increase in mechanical strength with decreasing fibre spacing did not seem to be linear. Additionally, an unpaired t-test comparing the means of yield stress between 40-layer/250 μm and multiscale scaffolds revealed a significant difference. This indicates that the incorporation of thinner fibres influences the yielding behaviour of the structures.

More importantly, the produced scaffolds were not able to match the mechanical properties of native acetabular labrum. The 40-layer/250 μm structures exhibited a mean Young’s modulus of 9.89 ± 2.26 Mpa, and multiscale structures exhibited values of 13.36 ± 2.59 MPa. In contrast, tensile modulus values for native acetabular labrum reported in literature were between 24.7 ± 10.8 MPa and 66.4 ± 42.2 MPa [19,20]. A predicted fibre spacing of 80 μm, using the multiple linear regression model, might be required for 40-layer structures to reach the tensile modulus of 24.7 MPa. According to the literature, however, for a 10-layer structure with fibre diameters of 15 μm, the minimum interfibre distance without introducing artifacts in the form of random fibre bridging to adjacent parallel fibres is reported to be 102 ± 2.7 μm, creating a design limitation in the current state of research [44]. Furthermore, the temporal evolution of mechanical properties through cell infiltration and extracellular matrix production was not considered in the present study.

Scaffold pore size influences cell adhesion (cell–scaffold interaction), cell–cell interaction, and cell migration across the scaffold, which is a fundamental part of tissue formation or regeneration [45,46]. The current study showed that primary labrum cells are able to directly migrate from the tissue specimen into the scaffolds. Migrated labrum cells as well as pre-cultured and seeded cells are both capable of attaching, spreading, and growing on the scaffolds (Figure 3). Cell–cell and cell–scaffold interactions could be observed on both scaffolds. For the explant scaffold co-culture study, it must be noted that these cells left their native cell niche, migrated out of the labrum tissue, fell due to gravity, and were then collected by the subjacent scaffold. Therefore, this study demonstrated cell compatibility with primary labrum cells but did not investigate guided cell migration into the scaffold. While tissue-engineering applications involve cell seeding and ex vivo growth of a labrum-like tissue, cell-free engineered grafts require cell migration to the reconstruction or repair site to create a repair tissue in vivo. One possible solution to attract present cells could be the functionalization of the scaffold surface with a growth factor, such as PDGF, which was recently shown to attract labrum progenitor cells to repair sites [47]. Cells on the grid-like structure (250/40) grew along the fibres and were not able to bridge the comparably large pores of 250 µm. Incorporation of fibres with smaller spacing and fibre diameter (multiscale) allowed the cells to occupy the space within the pores. Furthermore, multiscale scaffolds offered more possibilities for cell spreading, which were well accepted by the cells. Cells on a multiscale scaffold could still spread along the fibres and possess an elongated shape as did the cells on a grid-like scaffold (250/40); however, the majority of the cells made use of the present fibres in the near cell environment to spread out in space. Our findings showed that RGD-coated PCL melt electrowritten scaffolds are compatible with primary labrum cells and indicate that labrum cells prefer a multiscale architecture. Further research is required to investigate the effects of a multiscale architecture on the speed and quality of early tissue formation and on tissue remodeling towards a labrum-like tissue. It has been shown that allografts and autografts undergo a so-called “labralization”, a process during which the reconstructed graft tissue undergoes a biological transformation into labrum-like tissue [10,48,49].

The wave-patterned constructs reproduced the non-linear elastic behaviour of native acetabular labrum tissue [18]. The transition from the non-linear toe region to linear elastic region at 14.2 ± 1.8% strain allows the application of 10% cyclic tensile strain over 3600 cycles (1 h). Regarding the mechanical properties of native labrum tissue, 10% strain lies in the upper non-linear (toe region) or low linear elastic region of the stress–strain curve [18]. Recent studies suggested that local strains around 10% occur in the labrum tissue during daily activities [23,24]. With respect to construct elasticity, wave patterns might, therefore, be the preferred choice over grid-like PCL constructs to restore the mechanical function of the soft musculoskeletal tissue. However, it must be noted that 10% deformation of the construct (not to be mistaken for effective local material strains) occurred at lower loads when compared with the native labrum. Therefore, smaller fibre spacings are required to improve the mechanical strength. As mentioned above, however, smaller spacings increase the risk of fibre bridging [44]. Any irregularity in the manufacturing process drastically reduced the stretchability introduced by wave patterning. Another approach for the creation of highly elastic scaffolds by MEW has been published recently, where Diaz et al. developed a method that allows for the use of Poly(L-lactide-co-ε-caprolactone) (PLCL), a more elastic polymer than PCL, at a relative low temperature [50].

The overall metabolic activity was reduced in stretched compared with non-stretched scaffolds. One possible explanation could be a decreased cell number as a result of reduced proliferation and/or cell detachment from the scaffold under mechanical loading (Figure 4). It is important to note that 10% cyclic tensile strain at 1 Hz is considered physiological loading [23,24] and did not impact the metabolic activity of 2D cultured bovine labrum cells in a previous study [51]. However, in the case of wave-patterned scaffolds, the (hyper)elastic construct behaviour results from structural elasticity not material elasticity and, therefore, the cells did not experience a tensile strain of 10% locally. Instead, with each load cycle, substantial relative fibre movements occurred within the scaffolds to partially flatten the waves, potentially facilitating local cell detachment. Moreover, the large pore size of the scaffold provided limited possibilities for cell spreading and interconnection. With respect to cell compatibility, the use of a more elastic polymer, such as PLCL and a straight fibre architecture comprising fibres of lower fibre diameter and fibre spacing, might be a more promising alternative to restore the elasticity of the native tissue.

This study has potential limitations. While it is essential to understand the tensile properties of potential labral graft constructs, further research is required to determine the compressive properties, as well as the behaviour in more complex mechanical loading scenarios. The cell and tissue for the biological evaluation are sourced from bovine tissue and, therefore, do not fully reflect the human acetabular labrum cells and tissue with which a labral graft would interact. However, the high availability, easy access without burden, and homogeneity of the bovine model allow extensive basic research before translating into trials using human tissue.

## 5. Conclusions

This study provided a first evaluation of MEW scaffolds of varying architectures, with respect to the most relevant mechanical properties for labrum reconstruction. Moreover, a preliminary biological evaluation on primary labrum cell growth on MEW PCL scaffolds was performed for the first time. By treating MEW PCL scaffolds with RGD, it was shown that pre-cultured labrum cells as well as primary labrum cells migrating out of labrum explants into subjacent scaffolds can attach, spread, and grow on the scaffold structures, demonstrating cell compatibility and highlighting the potential as future labral graft candidates. Scaffold modulus, yield stress, and ultimate strength improve with decreased fibre spacing but are not influenced by the number of layers, implying that the determined properties are also valid for larger scaffold constructs. However, all the MEW PCL structures lacked the stiffness and strength of native tissue, achieving only 25–50% of the desired modulus value and less than 10% of the desired strength. Waved architecture scaffolds mimic the hyperelastic behaviour of the native tissue and allow for the application of physiological cyclic tensile strain; however, these did not enhance cell response. The use of a more elastic polymer and a straight fibre architecture might represent a more promising approach with respect to tensile strength and cell compatibility. As minimal fibre distance presents a natural limitation of melt electrowriting, impacts the precision, and, therefore, the elasticity of waved structures, a different thermoplastic should be considered to further develop melt-electrowritten labral grafts.

## Figures and Tables

**Figure 1 cells-11-03450-f001:**
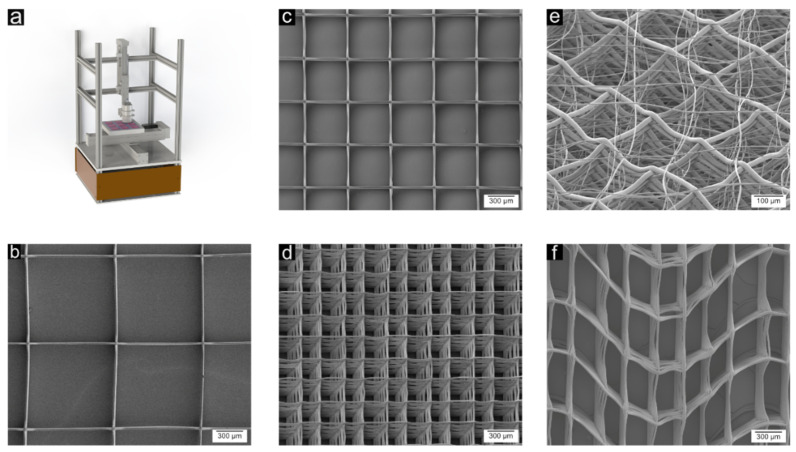
(**a**) CAD model of the custom-built MEW setup consisting of a melt head, XYZ axes, and a collector platform. (**b**–**d**) Top view of the manufactured PCL microfibre structures with 1000 µm, 500 µm, and 250 µm interfibre spacing. (**e**) Multiscale approach with defined grids with an interfibre spacing of 250 µm and a fine mesh of fibres alternating the grids. (**f**) Top view on waved structures for improved hyperelastic behaviour.

**Figure 2 cells-11-03450-f002:**
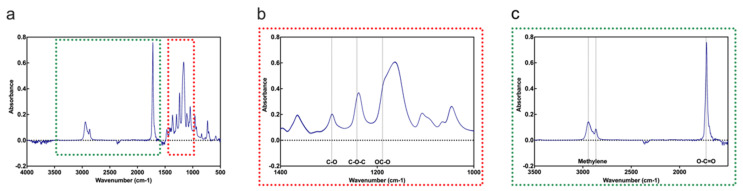
(**a**) The entire spectrum acquired using ATR-FTIR with two different focus areas marked and shown in (**b**,**c**). (**b**) Spectrum focusing on the characteristic absorption peaks around C-O, OC-O, and C-O-C bonds. (**c**) Spectrum focusing on the characteristic absorption peaks of methylene groups and ester carbonyl groups.

**Figure 3 cells-11-03450-f003:**
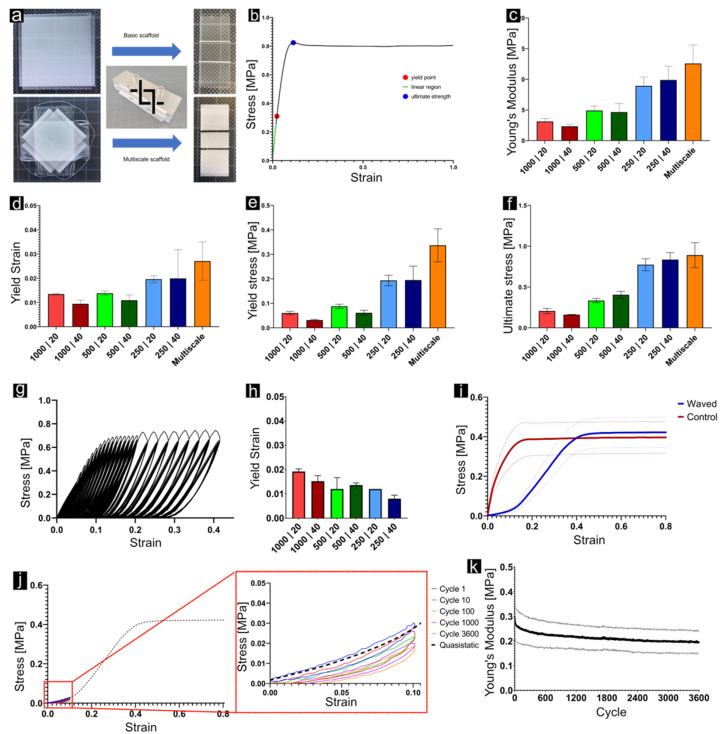
(**a**) Workflow for the preparation of test specimens for tensile tests, consisting of a custom-made blade holder for parallel cuts at a defined distance. (**b**) Exemplary stress–strain curve with yield point (red), linear region (green), and ultimate strength (blue). (**c**–**f**) Young’s modulus, yield strain, yield stress, and ultimate stress for each sample type. (**g**) Exemplary stress–strain curve for the stepwise cyclic test performed on structures with straight fibres. (**h**) Yield strain from each sample type derived from cyclic testing. (**i**) Tensile curves for the comparison of structures composed of waved against straight fibres. (**j**) Stress–strain curves from cyclic testing of waved structures up to 10% strain, well in the hyperelastic region. (**k**) Development of the modulus over the course of 3600 cycles during cyclic testing of waved samples.

**Figure 4 cells-11-03450-f004:**
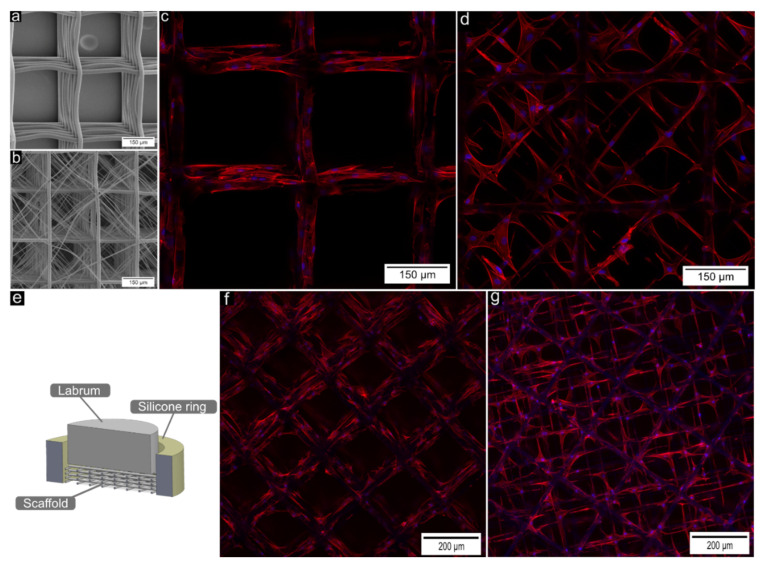
(**a**,**b**) SEM image of structures for labrum cell experiments with and without layers of fine fibres intermitting a defined grid of 250 µm interfibre spacing. (**c**,**d**) Confocal microscopy images of labrum cells (red: f-actin, blue: cell nuclei) growing on a grid-like scaffold without (**c**) and with (**d**) fibres of smaller diameter and smaller spacing incorporated. The scaffolds were seeded with cells and cultured for 2 days. In contrast with the grid-like scaffold, the multiscale scaffold (**d**) facilitates homogenous cell distribution in space and cell spreading. On both scaffolds however, high cell-material interaction can be observed, as indicated by the elongated/spread-out cell morphology. (**e**) Schematic representation of the co-culture experiment with a labrum piece sitting on top of a PCL scaffold with a porous silicone ring around. (**f**,**g**) Confocal microscopy images of labrum cells (red: f-actin, blue: cell nuclei) growing on a grid-like scaffold without (**f**) and with (**g**) fibres of smaller diameter and smaller spacing incorporated. The scaffolds were co-cultured with bovine labrum tissue specimen for 6 weeks. Both scaffold architectures allowed for cell migration and cell growth.

**Figure 5 cells-11-03450-f005:**
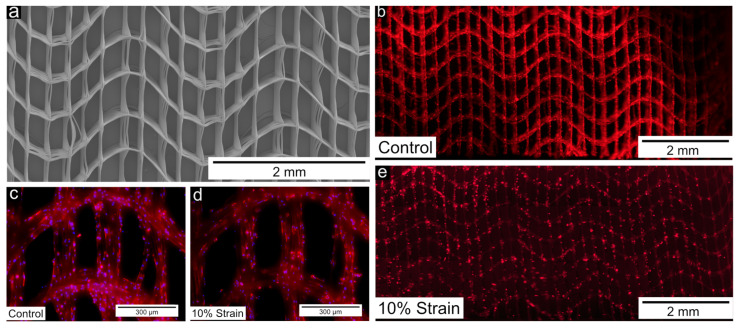
(**a**) Scanning electron microscopy image showing the architecture of wave-patterned scaffolds used for static and dynamic cell culture. (**b**–**e**) Fluorescent widefield microscopy images of actin-stained cells on wave-patterned scaffolds following long-term physiological stretching or static fixation (non-stretched control).

**Table 1 cells-11-03450-t001:** Overview of scaffold architectures (fibre spacing, wave amplitude, and layers) and applied experimental analysis (mechanical testing and cell culture study).

Sample	Fibre Apacing (Thin Fibres)	Wave Amplitude	Layers	Mechanical Testing	Cell Culture
1000/20	1000 µm	0 µm	20	quasi-static	
1000/40	1000 µm	0 µm	40	quasi-static	
500/20	500 µm	0 µm	20	quasi-static	
500/40	500 µm	0 µm	40	quasi-static	
250/20	250 µm	0 µm	20	quasi-static	
250/40	250 µm	0 µm	40	quasi-static cyclic	static
Multiscale	250 µm (100 µm)	0 µm	40	quasi-static cyclic	static
Wave 0	500 µm	0 µm	40	quasi-static cyclic	
Wave 500	500 µm	500 µm	40	quasi-static cyclic	dynamic vs. static

**Table 2 cells-11-03450-t002:** Values for Young’s modulus, ultimate strength, and yield stress (all in MPa) and yield strain (in %) for all straight fibre structures tested.

Sample	Young’s Modulus [MPa]	Ultimate Strength [MPa]	Yield Stress [MPa]	Yield Strain [%]
1000/20	3.12 ± 0.48	0.21 ± 0.03	0.06 ± 0.01	1.35 ± 0.03
1000/40	2.33 ± 0.34	0.16 ± 0.01	0.03 ± 0.003	0.94 ± 0.15
500/20	4.91 ± 0.71	0.33 ± 0.03	0.09 ± 0.01	1.38 ± 0.09
500/40	4.65 ± 1.42	0.41 ± 0.04	0.06 ± 0.01	1.09 ± 0.22
250/20	8.93 ± 1.46	0.77 ± 0.07	0.19 ± 0.02	1.96 ± 0.14
250/40	9.89 ± 2.26	0.84 ± 0.09	0.20 ± 0.06	1.99 ± 1.19
Multiscale	13.36 ± 2.59	0.90 ± 0.17	0.33 ± 0.07	2.39 ± 0.17

## Data Availability

The data presented in this study are openly available in the ETH Zürich Research Collection at 10.3929/ethz-b-000573422.

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
