# Peer review of "Mechanical and Biological Evaluation of Melt-Electrowritten Polycaprolactone Scaffolds for Acetabular Labrum Restoration"

_cells, 2022, doi:10.3390/cells11213450_

Round 1
Reviewer 1 Report
This study entitled “Melt-electrowritten polycaprolactone scaffolds for acetabular labrum restoration” seems to be well executed and written paper. Furthermore, I believe that this paper will be of great interest to the readers. Finally, I have only a few suggestions to further improve the quality of this interesting paper.
Title
Please include the type of study that you have performed in your title.
Introduction
Please state the clear hypothesis of your research at the end of Introduction section.
Materials and Methods
Please begin this section with an information what type of study you have performed, in which time period and where.
Data analysis
What type of multiple linear regression you have performed (forward, backward, enter,…). Please include that information in your text.
Discussion
Please state the limitations of your study at the end of Discussion section.
Author Response
Thank you very much for your review, which helped to improve the manuscript.
Point 1: Please include the type of study that you have performed in your title.
Response 1: The title has been adapted to “Mechanical and biological evaluation of melt-electrowritten polycaprolactone scaffolds for acetabular labrum restoration”
Point 2: Please state the clear hypothesis of your research at the end of Introduction section.
Response 2: The hypothesis of the research is stated at the end of the Introduction section: “Hypothesis”- Have to turn the following sentence into a hypothesis or add a hypothesis in the end: “The goal of this study was to compare melt-electrowritten PCL scaffolds of varying architectures with respect to most relevant mechanical properties and identify ideal candidates for labrum reconstruction. Moreover, a preliminary biological evaluation was conducted to provide insights on primary labrum cell growth. We hypothesised that: (i) mechanical properties improve with decreased fibre spacing are independent of construct thickness, (ii) hierarchichal scaffolds improve cell spreading without altering mechanical properties and (iii) biomimetic wave-patterned scaffolds reproduce the non-linear mechanical response of labrum tissue and enhance cell response.”
Point 3: Please begin this section with an information what type of study you have performed, in which time period and where.
Response 3: The information about the study type, time period and location has been added to the beginning of the section “Materials & Methods”: “Overview. In this study, PCL constructs with a range of architectures were manufactured using MEW. The fabricated constructs were structurally assessed using scanning electron microscropy, mechanically by means of quasistatic and cyclic uniaxial tensile testing and biologically using static and dynamic cell culture with primary bovine acetabular labrum cells as well as static tissue co-culture with primary bovine acetabular labrum explants. The experiments have been conducted at the Laboratory for Orthopaedic Technology of ETH Zurich over a time frame of six months.”
Point 4: What type of multiple linear regression you have performed (forward, backward, enter,…). Please include that information in your text.
Response 4: In the present study, multiple linear regression has been used to understand which process variables influence the dependent outcome variable, with no stepwise selection (i.e. enter method). In the manuscript, we have now tried to point this out more clearly.
Point 5: Please state the limitations of your study at the end of Discussion section.
Response 5: A statement discussing the limitation of the current study has been added to the end of the Discussion section: “This study has potential limitations. While the tensile properties of potential labral graft constructs are essential to understand, further research is required to determine the compressive properties, as well as the behavior in more complex mechanical loading scenarios. The cell and tissue for the biological evaluation are sourced from bovine tissue and therefore do not fully reflect the human acetabular labrum cells and tissue a labral graft would interact with. However, the high availability, easy access without burden, and homogeneity of the bovine model allow extensive basic research before translating into trials using human tissue.”
Reviewer 2 Report
Reviewer’s comments:
The manuscript entitled ‘Melt-electrowritten polycaprolactone scaffolds for acetabular labrum restoration’ has been peer-reviewed. The authors have characterized the structure and tensile mechanical properties of melt-electrowritten polycaprolactone scaffolds of varying architectures and determined their cell viability for the reconstruction of the labral tissue. We have provided the following comments to improve the manuscript.
Minor concerns
1) The abstract should end with a concluding remark.
2) Though the polymeric substrate (PCL) is well-known, it should be characterized by any spectral data like ATR-FTIR.
Author Response
Thank you very much for your helpful review, which improved our manuscript.
Point 1: The abstract should end with a concluding remark.
Response 1: The abstract has been adapted to end with a concluding remark: “In conclusion, melt-electrowritten polycaprolactone scaffolds are promising candidates for labral grafts, however further development is required to improve both the mechanical and biological compatibility.”
Point 2: Though the polymeric substrate (PCL) is well-known, it should be characterized by any spectral data like ATR-FTIR.
Response 2: The spectral data analyzing the polymeric substrate has been added to the methods and results section and discussed accordingly.